# Implication of COVID-19 Pandemic on Adolescent Mental Health: An Analysis of the Psychiatric Counseling from the Emergency Room of an Italian University Hospital in the Years 2019–2021

**DOI:** 10.3390/jcm11206177

**Published:** 2022-10-19

**Authors:** Maria Giuseppina Petruzzelli, Flora Furente, Giuseppe Colacicco, Federica Annecchini, Anna Margari, Alessandra Gabellone, Lucia Margari, Emilia Matera

**Affiliations:** 1Department of Basic Medical Sciences, Neuroscience and Sensory Organs, University Hospital “A. Moro”, Piazza Giulio Cesare 11, 70100 Bari, Italy; 2Department of Biomedical Sciences and Human Oncology, University Hospital “A. Moro”, Piazza Giulio Cesare 11, 70100 Bari, Italy

**Keywords:** COVID-19 pandemic, adolescents’ mental health, self-harm

## Abstract

Introduction: Although the COVID-19 pandemic had profound consequences on youths’ mental health, few data are available about its longitudinal implications. Method: In this study, from 655 counseling requests by the Emergency Room (ER) of the University Hospital of Bari, we retrospectively examined 380 requests for psychiatric counseling of pediatric subjects, during the pre-pandemic, the first pandemic, and the second pandemic wave of COVID-19. Results: We found a significant upward trend between 2019 and 2021 for the counseling requests for acute psychopathological symptoms (*p* = 1.469 × 10^−5^), patients in adolescent age (*p* = 0.022), females (*p* = 0.004), and those taking psychotropic medications (*p* = 2.28 × 10^−5^). Moreover, a significant difference in the proportions of depression (*p* = 0.003), post traumatic (*p* = 0.047), somatic (*p* = 0.007) and psychotic symptoms (*p* = 0.048), and self-injuring behaviors (*p* = 0.044) was observed. The proportion of counseling for psychotic symptoms (*p* = 0.014) and self-injuring behaviors (*p* = 0.035) also showed an increasing trend over time, with self-harming behaviors becoming more severe and diversified in modalities. Discussion: The pandemic’s persistence over time may have had an impact on youth’s psychopathology, influencing the frequency, type, and complexity of mental health problems; as a result, it is vital to implement timely integrated interventions and find strategies to prevent self-harm, in particular with the identification of vulnerable categories of patients.

## 1. Introduction

On 11 March 2020, the World Health Organization (WHO) declared COVID-19, a respiratory disease caused by the SARS-CoV-2 virus, to be a pandemic and an international health emergency [1]. As a result, governments around the world implemented containment measures, including closing schools and non-essential businesses, promoting working from home and mass quarantine, and discouraging international travel, which quickly affected many aspects of millions of people’s lives around the world. The social distancing and isolation, the increased feeling of financial death and disease risk, the lack of necessary supplies and knowledge, and the loss and stigma seemed to have a profound impact on people’s mental health and on the exacerbation of any pre-existing mental health problems [2,3]. Various groups of people, including children and adolescents, as well as pregnant women, homeless people, migrants, people at risk of domestic abuse, lower socio-economic groups, and people with a history of mental health, were found to be more vulnerable to the mental health implications of the COVID-19 pandemic [4]. In general, children and adolescents appear to be more sensitive to the effects of COVID-19 than adults, which may contribute to symptoms like anxiety, depression, distraction, irritability, and sleep disturbances [4,5,6]. However, prior to the COVID-19 pandemic, alarming data on the mental health of the youth population had already been made available to the public. According to the data published by the National Institute of Mental Health (NIMH), among US adolescents, the prevalence of any mental disorder was approximately 49.5%, [7]. Moreover, the At-Risk Behavior Survey (YRBS) from the Centers for Disease Control and Prevention revealed a significant change in the percentage of occurrences of self-injurious behavior, an increasing issue in young people’s mental health, between 2009 and 2019 [8].

The majority of the explanatory models of suicidal behaviors emphasize the interaction between predisposing and precipitating factors via the stress dimension [9]. According to the stress vulnerability or stress-diathesis model, the protraction of the pandemic over time may be considered an important stressful factor leading to immediate and long-term mental health consequences on children and adolescents, given the fragile condition of adolescent’s mental health prior to COVID-19 [10]. Recent studies conducted during the first COVID-19 pandemic wave have already indicated a dramatic rise in the prevalence of mental illness in children and adolescents worldwide, particularly in relation to anxiety, depression, substance use, and self-harm [11,12,13]. Otherwise, considering that the pandemic is still ongoing, no data is available on the long-term implications of the COVID-19 pandemic on the mental health of children and adolescents.

Although it is quite intuitive that an emergency such as a pandemic causes an increase in acute psychopathological distress and that adaptation to a prolonged critical period depends on resilience and coping strategies to continue to function well despite adverse circumstances [14], it is important to examine the impact of the persistence of the pandemic on the mental health in young people. To date, several studies have focused on psychiatric emergencies during the COVID-19 pandemic and analyzed requests for psychiatric counseling in the Emergency Room (ER), but these were studies of adult populations [15,16,17,18,19,20].

Our study hypothesized that the persistence of this stressful event might have an impact on the psychopathological conditions of young people, as well as differences in the nature and severity of these conditions, especially between the first and second waves of the COVID-19 pandemic. It is worth noting that Italy, with 16 million infections and over 160 thousand deaths related to the infection COVID-19 between March 2020 and April 2022, was among the European countries most affected by the pandemic, especially in the first phase, with a significant improvement in 2021 coinciding with the start of the vaccination campaign [21]. In addition, Italian data on the psychiatric impact of the persistence of the pandemic refer only to the period immediately after this event [17].

Based on these premises, the main aim of the study was to evaluate the requests for psychiatric counseling from the Emergency Room (ER) of the University Hospital of Bari, Italy, in subjects younger than 18 years old, comparing the periods pre-pandemic (from 1st July to 31st December 2019, eight to two months before the start of the COVID-19 pandemic), the first COVID-19 wave (from 1 July to 31 December 2020, four to nine months after the start of the COVID-19 pandemic), and the second COVID-19 wave (from 1 July to 31 December 2021, sixteen to twenty-one months after the start of the COVID-19 pandemic). Specifically, we focused on the following: (1) the rate/percentage of requests for acute psychopathological symptoms (APSr) on the total counseling requests from ER; (2) the types of psychopathological symptoms motivating the presentation to the ER (anxiety, agitation, eating behaviors, depression, somatization, trauma, psychosis, drug abuse, self-harm); (3) the evaluation of increasing trends across the different periods of the COVID-19 pandemic (pre-pandemic, post first-wave, post second-wave).

## 2. Materials and Methods

### 2.1. Design

This study is a retrospective assessment of the trend of acute psychopathological symptoms in three time periods (pre-pandemic, COVID-19 first and second waves).

### 2.2. Participants

Participants were children and adolescents up to 18 years of age who required urgent specialist counseling from the Emergency Room (ER) of the University Hospital of Bari for the onset of APS (anxiety, agitation, eating behaviors, depression, somatization, trauma, psychosis, drug abuse, self-harm) in the pre-pandemic, COVID-19 first and COVID-19 second wave. One parent/legal guardian was always present during the counseling.

### 2.3. Procedure

A retrospective analysis of the counseling requests was performed using electronic medical records from the hospital computerized operating system, “Galileo” that is used by doctors, nurses, and management personnel with personal credentials. It provides and collects information relating to patients’ identification, reception, clinical and diagnostic procedures, and results that are shared by all the hospital’s clinics, guaranteeing a fully integrated management. Consultations performed were included in the pre-pandemic, first wave, and second wave groups. Gathered data were age, gender (female gender), and previous prescription of psychotropic drugs (previous drugs). According to the evaluation by a child neuropsychiatrist on the basis of clinical observation, declared symptoms, and 5th edition of Diagnostic and Statistical Manual of Mental Disorders (DSM-5) diagnostic criteria [22], the main symptoms motiving the requests for counseling were grouped in different psychopathological dimensions including anxiety, behavioral agitation (agitation), depressive symptoms (depression), eating behaviors, somatic symptoms (somatization), psychotic symptoms (psychosis), traumatic/stressful event-related symptoms (trauma), substance use related symptoms (drug abuse), and self-injuring behaviors (self-harm), for both suicide attempts and non-suicidal self-injury. All the above data were independently screened and systematized by five authors (F.F., F.A., G.C., A.M. and A.G.). For the management of any uncertainties and errors on the classification of the reasons for consulting, the data collected were cross-checked by an expert child neuropsychiatrist (M.G.P. and E.M.).

The study was approved by Local Independent Ethics Committee of the Policlinic od Bari (PS-C19).

### 2.4. Setting

We included all the requests for counseling from the Central ER, the Ophthalmological ER, and the ER of the Giovanni XXIII pediatric hospital; requests for which any data or consulting reports were missing were excluded, as well as requests for reasons other than APSr (i.e., epilepsy, migraine, multiple sclerosis, accidental head traumas, etc.). Moreover, non-urgent counseling requests from other units were excluded too.

### 2.5. Data Analyses

All the variables were collected in a structured form, specific for this research. Descriptive and statistical analyses were performed using the R statistical environment, version 4.1.2 (The R Foundation for Statistical Computing; Vienna, Austria). Continuous variables are reported in means and standard deviations, and categorical variables are expressed in numbers and percentages. For categorical data, independence of the proportions was verified by the Pearson-X^2^ test. Lastly, Mantel–Haenszel linear-by-linear association X^2^ test investigated the alternative hypothesis that there is a linear association between the row variables and the column ones. *p*-values lower than 0.05 were considered statistically significant. A Shapiro–Wilk test and graphical evaluations of continuous variables were performed to demonstrate the eventual correspondence with the normal distribution. Significant differences in continuous variables were analyzed by a Kruskal–Wallis test, followed by pairwise comparisons using Wilcoxon rank sum test with continuity correction. The one-sided Jonckheere–Terpstra test for ordered alternatives investigated the alternative hypothesis of an upward trend for median of continuous variables.

## 3. Results

Table 1 summarizes all the results on the socio-demographic and clinical variables that we evaluated. We analyzed a total number of 655 counseling requests; 6 of them were excluded for lack of report and 269 were excluded for reasons other than APSr (i.e., epilepsy, migraine, multiple sclerosis, accidental head traumas, etc.). It should be noted that since the selected cases were registered in the same hospital center, there was a possibility that some of them had asked for help more than once, and that there may be multiple reports for the same patients at multiple points in time. Ultimately, 246 consultations were from the pre-pandemic group, 178 from the first wave group, and 225 from second wave group. After evaluation of the motivation of counseling requests, we found a final number of 125 consultations for APSr in pre-pandemic group, 96 in the first wave group, and 159 in the second wave group. The mean ages were 13.3 years (SD ± 3.65) in pre-pandemic group, 12.8 years (SD ± 4.15) in the first wave, and 14.0 years (SD ± 3.42) in the second wave group, with a significant difference between the three groups (*p* = 0.039). The Jonckheere–Terpstra test for ordered alternatives showed an upward trend for age among such groups (*p* = 0.022).

The Pearson-X^2^ test showed that, when comparing data between the three periods, there were significant differences between the proportion of requests for APSr (*p* = 2.439 × 10^−5^), the proportion of female gender (*p* = 0.009) and the proportion reporting previous drugs (*p* = 7.733 × 10^−5^) on the total of requests. Moreover, the Mantel–Haenszel-X^2^ test, performed on the same proportions, showed a significant linear increasing trend for all of them (APSr *p* = 1.469 × 10^−5^, female gender *p* = 0.004, previous drugs *p* = 2.28 × 10^−5^).

Considering the comparison of psychopathological dimensions between the three periods, the Pearson-X^2^ test showed significant differences between the proportion of requests for depression (*p* = 0.003), somatization (*p* = 0.007), trauma (*p* = 0.047), psychosis (*p* = 0.048), and self-harm (*p* = 0.044) on the total of APSr; no significant differences were found for the proportion of requests for counseling related to anxiety (*p* = 0.203), agitation (*p* = 0.545), eating behaviors (*p* = 0.115), and drug abuse (*p* = 0.268).

The Mantel–Haenszel-X^2^ test performed on the same proportions indicated significant increasing trend only for psychosis (*p* = 0.014) and self-harm (*p* = 0.035). Moreover, Figure 1 illustrates the reciprocal ratio between suicide attempts and self-injuring behaviors, Figure 2 shows the reported injury modalities during the three examined periods (pre-pandemic, COVID-19 first wave, COVID-19 second wave).

## 4. Discussion

The main findings of this study were that there was a significant upward trend in the percentage of counseling requests for acute psychopathological symptoms and differences in the type and severity of these conditions when comparing the three-year period 2019–2021, especially between the first and second waves of the pandemic COVID-19, as detailed below.

### 4.1. Rate/Percentage of Requests for Acute Psychopathological Symptoms

During the first wave of COVID-19 pandemic, there was a slight decrease in the total absolute number of counseling for APSr to the ER of the University Hospital of Bari, followed by a subsequent increase during the second pandemic wave. More specifically, in the three-year period 2019–2021, the percentage of demands for APSr support exceeded the demands for neurological issue support, with a significant increasing trend. The absolute number and percentage of counseling for patients of adolescent age, female gender, who were already receiving treatment with psychotropic medicines managed by outpatient specialist services showed similar trends.

Several factors, such as social distancing and isolation, reduced travels, and the choice of sick and suffering people to stay at home out of fear of contracting the disease, could explain the decline of the ER visits during the first wave of COVID-19 pandemic [23]. The increase in ER visits for APSr during this time period may be related to the decreased availability of outpatient specialist care and to the not-in-person alternatives for the management of diseases (by phone or video call) used by doctors; their further increase, especially during the COVID-19 second wave of the pandemic, when specialist outpatient services had been gradually restored, reinforces the theory that the persistence in time of the COVID-19 pandemic had a damaging effect on youth psychic well-being, with plausible long term implications, especially on those with poor mental health prior to the pandemic [24]. This condition has inevitably heightened the demand for therapeutic intervention programs, which often involve the introduction of a pharmacological therapy, intensifying a tendency already developing in previous years. In fact, Italy in 2020, the prevalence in the use of psychotropic drugs (in particular antipsychotics and antidepressants) in young people was already 0.3% with a prescription rate of 28.2 per 1000 youths, with a rise of 11% compared to the previous year [25].

In general, age-specific features may expose adolescents to a greater risk of mental health difficulties than the adult population [26]. Even in the best environmental condition, adolescence is known as a period of greater vulnerability due to a variety of factors, such as physical development, brain chemical changes that intensify emotional reactions to stress-inducing factors, a slower development of the emotional self-regulation system, and increased social sensitivity. A large meta-analysis of 29 studies involving more than 80,000 young people showed older children were more likely than younger children to experience symptoms of anxiety, depression, and an unfulfilled life, because they were more affected by restrictions that prevented them from sharing experiences and important growth milestones with their peers, such as first relationships and final exams, and being forced to stay at home led them to arguments with their parents in a moment of life in which autonomy is deeply sought [27,28]. Female adolescents, who are typically more at risk of psychopathology than male adolescents, appear to be more sensitive to these factors because they are more prone to rely on social support in stressful situations [29].

### 4.2. Types of Psychopathological Symptoms Motivating the Presentation of the ER and Evaluation of Increasing Trends through the Different Period of COVID-19 Pandemic

Between 2019 (pre pandemic) and 2021 (second pandemic wave), a significant difference in the proportions of depression, post-traumatic, somatic, and psychotic symptoms and self-injuring behaviors was recorded. Specifically, if the requests of counseling for depressive symptoms decreased in 2020 and returned to pre-pandemic levels in 2021, those for traumatic and somatization symptoms (gastrointestinal complaints such as colitis, diarrhea, nausea and vomiting, headache, back and widespread pain) increased during COVID-19 first wave to return to pre-pandemic levels in 2021, as already reported by previous studies. During the pandemic, compared to the pre-COVID-19 period, the incidence rates of stress-sensitive gastrointestinal diseases such as ulcerative colitis, Crohn’s disease, and esophageal reflux increased significantly, with some exceptions. Cases of irritable bowel syndrome, in fact, were likely reduced due to blockages linked to the pandemic (access to domestic sanitation facilities, adoption of flexible work and school hours, improvements in sleep and physical exercise, and a sense of psychophysical well-being) [30]. Moreover, the proportion of urgent counseling requests for psychotic symptoms and self-injuring behaviors revealed an upward trend over time, with a peak during the second wave of COVID-19. If we focus on self-harming, during the three years 2019–2021, the number of suicide attempts (SA) compared to the NSSI progressively increased, as well as the modalities of injury: cutting remained the most frequent form of NSSI, but in 2021 cases of pinching were also recorded; attempted suicide, which in pre-pandemic periods came about predominantly using blades, in 2020 and 2021 occurred through drugs and toxics-ingestion.

These findings lead us to hypothesize that the psychological reaction to the pandemic crisis was different during the first and second wave of COVID-19. Anxious reactions predominated during the first wave, which in young people led to somatization and post-traumatic discomfort, validating the points made by several authors who concentrated on this subject throughout this period of pandemic [4,5,6]. During the second pandemic wave the psychological distress was manifested with depressive symptoms. Living with restrictions and social distancing, with more frequent parental conflicts, and with a significant increase in the number of hours young people spent online and on social media dramatically reducing the positive effect provided by peers, teachers, and community connections for personal, emotional and social development, may have led to increasing feelings of loneliness [27,31]. A recent meta-analysis suggested that social isolation and loneliness could predict future mental health issues and could be linked to mental health issues up to 9 years later, and their duration appeared to be predictors of future mental health problems [28].

The number of counseling requests for APSr surged during the second wave of the COVID-19 pandemic, with an increase in clinical and etiopathogenetic complexity, including psychosis symptoms and self-harming behavior. However, therapeutic interventions for self-harm are not as immediate and direct as other conditions, for example psychosis, for which specific pharmacological treatments also exist for pediatric patients.

Prior research indicated that self-harm and SA decreased in the early months of the COVID-19 pandemic but then increased to pre-lockdown frequencies at the end of follow up, while other research found an increase in these behaviors during the same time period [32] without any acknowledgment of the subsequent phase. According to the Substance Abuse and Mental Health Services Administration, it is known that self-harming rates decrease during the initial phase (“honeymoon”) of a crisis; but this phase is generally followed by the “disillusionment” due to the limitations of assistance and feelings of abandonment, and this can last for a long time [33] that is usually more severe [34,35,36,37,38], particularly for individuals with pre-existing psychiatric conditions [39,40,41]; our data was supported by this theory.

Suicidality and self-harm are two urgent public health problems that are difficult to prevent and are closely interrelated: self-injuring behaviors are risk factors for SA, and previous episodes of self-injuring behaviors and suicidality are considered predictors of further self-harming behaviors over time [39,42,43,44,45]. SA and self-injuring activities are alarming phenomena also because the youth population is severely impacted. In addition to the known rapid physical and social changes occurring during adolescence, structural and functional alterations of the brain in this period of life (i.e., amygdala circuits, white matter microstructure in the uncinate fasciculus, cingulum, bilateral superior and inferior longitudinal fasciculi, anterior thalamic radiation, callosal body, and corticospinal tract) could represent a possible factor of neurobiological vulnerability with regard to the development of maladaptive coping strategies [46,47,48]. The gender gap in self-harming behaviors’ incidence, higher in females, could be due to the known sex-related difference on the incidence of internalizing problems, already noticeable in early adolescence, to the sex-specific hormonal differences in the estrogen-testosterone ratio, and to the variability in the timing of physical development and changes associated with menarche and menstruation occurring through the modulation of the neuroendocrine system [26,49,50,51].

The greater variety of self-harming manifestations is a further element of concern and challenge. Already in 2020 in France there was an increase in suicide attempts using more lethal methods such as firearms, jumping, and drowning compared with the same period in 2019 (from January to August) [52]. However, for this aspect we do not have data relating to the subsequent period.

### 4.3. Limitations of the Study

There are some limitations in this study, including the small sample size and the problems related to the measurement of psychiatric symptoms in an ER setting. Since the data was collected from computer systems and referred to first aid consultations, its accuracy was influenced by the quality of the information entered in the archives. Due to these reasons, we have not examined additional risk factors for mental illness (e.g., family conflicts, familiarity, media exposure); moreover, the subsequent course taken by patients is unknown, because they have not undergone to an evaluation for psychopathology. As a result, it is likely that the frequency of psychopathological symptoms in the sample was underestimated. Finally, data from this study did not cover the whole pandemic period and should thus be considered preliminary.

## 5. Conclusions

The persistence of the COVID-19 pandemic over time has led to an increase in the number of urgent counseling requests for APSr, in particular for certain categories of patients (adolescent age, female gender, already being treated with psychotropic drugs). In addition, between 2019 (pre-pandemic) and 2021 (the second pandemic wave), significant differences emerged in the proportions of APSr including depression, post-traumatic symptoms, and somatizations, and, in particular, an upward trend over time of severe and complex symptoms such as psychosis and self-harm. It is therefore urgent that timely interventions are put in place to promote and protect the mental health of young people, to define measures that in particular prevent self-harm, to involve the government and non-governmental organizations to improve access to psychiatric care, and to enhance different levels of assistance such as digital media, health and social structures (clergy, school workers, and health workers) that are educated in mental illness [10,11,53].

Finally, further longitudinal research is needed to define the age and characteristics of groups with particular vulnerability to mental health implications of the COVID-19 pandemic and other similar emergency situations, and which are a starting point for developing targeted mental health management strategies.

## Figures and Tables

**Figure 1 jcm-11-06177-f001:**
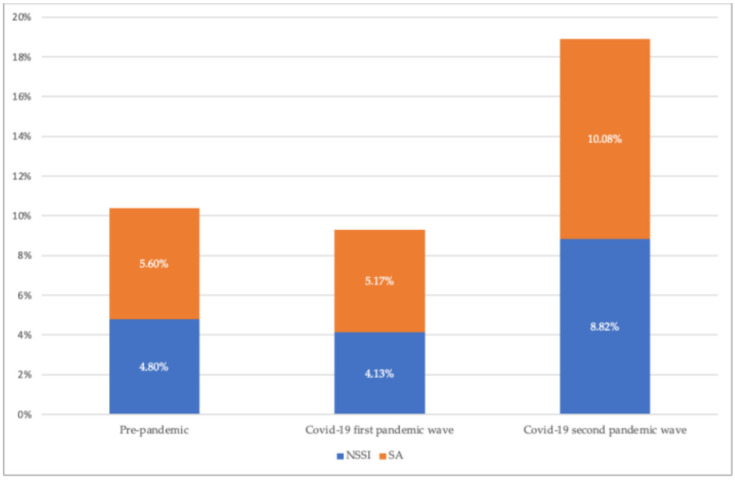
Reciprocal ratio between SA and NSSI.

**Figure 2 jcm-11-06177-f002:**
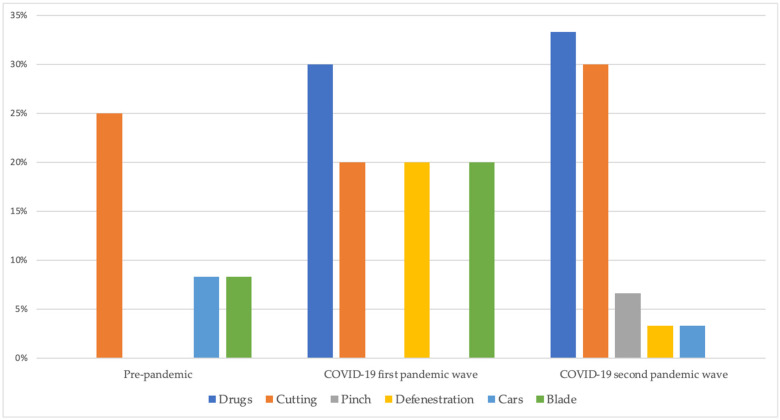
Proportion of injury modalities during the three examined periods.

**Table 1 jcm-11-06177-t001:** Summarizes all the results about sociodemographic and clinical variables.

Sociodemographic and Clinical Variables								
	Pre-Pandemic		COVID-19 First Pandemic Wave		COVID-19 Second Pandemic Wave		*p*-Value	
	Mean (SD)	Median	Mean (SD)	Median	Mean (SD)	Median	Kruskal–Wallis	Jonckheere–Terpstra
Age	13.3(3.65)	14.0	13.3(3.65)	14.0	14.0(3.42)	15.0	0.039	0.022
	Pre-pandemic		COVID-19 first pandemic wave		COVID-19 second pandemic wave			
	*n* = 246		*n* = 178		*n* = 225		Pearson-X^2^	Mantel–Haenszel-X^2^
APSr *n* (%)	125 (50.8%)		96 (53.9%)		159 (70.7%)		2.439 × 10^−5^	1.469 × 10^−5^
	*n* = 125		*n* = 96		*n* = 159		Pearson-X^2^	Mantel–Haenszel-X^2^
Female gender *n* (%)	56 (44.8%)		46 (47.4%)		98 (61.6%)		0.009	0.004
Previous drugs *n* (%)	32 (25.6%)		31 (31.9%)		79 (49.7%)		7.733 × 10^−5^	2.28 × 10^−5^
Psychopathological Dimensions								
	Pre-pandemic		COVID-19 first pandemic wave		COVID-19 second pandemic wave			
	*n* = 125		*n* = 96		*n* = 159		Pearson-X^2^	Mantel–Haenszel-X^2^
Anxiety *n* (%)	53 (42.4%)		30 (30.9%)		57 (35.8%)		0.203	0.293
Agitation *n* (%)	48 (38.4%)		39 (40.2%)		71 (44.7%)		0.545	0.281
Eating behaviors *n* (%)	15 (12.0%)		4 (4.1%)		16 (10.0%)		0.115	0.661
Depression *n* (%)	19 (15.2%)		2 (2.0%)		24 (15.0%)		0.003	0.856
Somatization *n* (%)	12 (9.6%)		20 (20.6%)		13 (8.2%)		0.007	0.577
Trauma *n* (%)	6 (4.8%)		10 (10.3%)		5 (3.1%)		0.047	0.453
Psychosis *n* (%)	5 (4.0%)		7 (7.2%)		19 (11.9%)		0.048	0.014
Drug Abuse *n* (%)	2 (1.6%)		4 (4.1%)		2 (1.3%)		0.268	0.766
Self-harm *n* (%)	13 (10.4%)		9 (9.3%)		30 (18.9%)		0.044	0.035

## Data Availability

The data presented in this study are available on request from the corresponding author.

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
