# Peer review of "Implication of COVID-19 Pandemic on Adolescent Mental Health: An Analysis of the Psychiatric Counseling from the Emergency Room of an Italian University Hospital in the Years 2019–2021"

_jcm, 2022, doi:10.3390/jcm11206177_

Round 1
Reviewer 1 Report (New Reviewer)
Dear authors
My congratulations for a good job in which a robust and up-to-date background, adequate methodology, and results and discussion. well elaborated, but I suggest some recommendations:
A more complete description regarding the IMRaD structure in the abstract. It is recommended to make the aforementioned structure transparent.
Although the methodological design is contingent, a greater characterization of the sample would be recommended.
Draw more specific conclusions
Best regards
Author Response
Dear reviewer,
we thank you for your helpful suggestions. Here below you will find the answers to your requests that we hope to have satisfied.
A more complete description regarding the IMRaD structure in the abstract. It is recommended to make the aforementioned structure transparent.
As you suggested, we made the abstract IMRaD structure more transparent.
Although the methodological design is contingent, a greater characterization of the sample would be recommended.
In the Material and Methods section (Participants) of the manuscript we better characterized the sample of the study (2.2 Participants)
Draw more specific conclusions
As you requested, we drew more specific conclusions
Best regards,
Emilia Matera

Reviewer 2 Report (New Reviewer)
The description of the sample, primarily the exclusion/inclusion criteria is not clear: initially, the authors identified 246 consultations in the pre-pandemic group, but only 125 cases entered the analyses; 178 from in the COVID-19 first pandemic wave group – but only 96 cases entered the analyses and 225 from the COVID-19 second pandemic wave group – 159 of which were left in the analyses. The authors state that these were the numbers “after evaluation of the motivation of counseling requests” but do not describe how exactly did this process of evaluation of motivation look like? Previously (in the participants section), it was stated that all of these patients required specialist counseling for the onset of acute psychiatric symptoms. So, patients with which symptoms (and why?) were excluded?
Furthermore, since all the cases were registered in the same hospital center, there is a possibility that some of them sought help more than once, and that there are multiple reports for the same patients (in more than one time points of this study): was this taken into account?
Since the data were drawn from medical records from the hospital computerized operating system (“Galileo”), would it be possible to retrieve the data regarding their COVID status as well? It would be informative to check is certain symptoms are more frequent post-infection (given the rising number of studies showing neurological impairments in adolescents in months following the COVID infection).
Author Response
Dear Reviewer,
we thank you for your considerations and for the suggestion of changes that will certainly be useful to improve both the quality of our manuscript and our future studies. Below we list the responses to your reviews, which we hope can be considered satisfactory by you.
The description of the sample, primarily the exclusion/inclusion criteria is not clear: initially, the authors identified 246 consultations in the pre-pandemic group, but only 125 cases entered the analyses; 178 from in the COVID-19 first pandemic wave group – but only 96 cases entered the analyses and 225 from the COVID-19 second pandemic wave group – 159 of which were left in the analyses. The authors state that these were the numbers “after evaluation of the motivation of counseling requests” but do not describe how exactly did this process of evaluation of motivation look like? Previously (in the participants section), it was stated that all of these patients required specialist counseling for the onset of acute psychiatric symptoms. So, patients with which symptoms (and why?) were excluded?
As you suggested, in the Results section we better described the exclusion/inclusion criteria for the sample of the study.
Furthermore, since all the cases were registered in the same hospital center, there is a possibility that some of them sought help more than once, and that there are multiple reports for the same patients (in more than one time points of this study): was this taken into account?
As you rightly pointed out, in the Results section we included a sentence about the possibility that some of the enrolled patients have sought help more than once and that there have been multiple reports for the same patients at more than one point in our study.
Since the data were drawn from medical records from the hospital computerized operating system (“Galileo”), would it be possible to retrieve the data regarding their COVID status as well? It would be informative to check is certain symptoms are more frequent post-infection (given the rising number of studies showing neurological impairments in adolescents in months following the COVID infection).
We agree on the usefulness of verifying whether some psychiatric symptoms are more frequent after COVID infection. Although the study data were obtained from the medical records of the hospital's computer system, we used the section relating to first aid consultations which do not report any further information other than that already reported. We are sorry not to be able to enrich the manuscript by using your helpful suggestion.
Best regards,
Emilia Matera

This manuscript is a resubmission of an earlier submission. The following is a list of the peer review reports and author responses from that submission.
Round 1
Reviewer 1 Report
Dear Authors,
Thank you for the opportunity to review the paper entitled “Longitudinal implication of Covid-19 Pandemic on adolescent mental health: a retrospective analysis in the years 2019-2021 of the psychiatric counseling from the Emergency Room of the University Hospital of Bari, Italy”. The study examined the requests for psychiatric counseling during pandemic wives and compared this to the pre-pandemic period. I find the content of this manuscript challenging. My first impression is that the type of article indicated incorrectly, it should be communication. My overall impression reveals a low scientific level of the study. The findings alone, that the number of people seeking counseling increased during the pandemic is obvious (we know this from numerous meta-analyses) - it doesn't add anything new to the science, other than to point out that this was also the case at the hospital in Bari. In this form, it is hard to find a strong element of the study. Below is a list of my comments:
Major:
· Is it a cross-sectional study design?
· Chapter 2.1 is far too short - please expand with detailed descriptions of data acquisition methods
Minor:
· I would suggest presenting the last paragraph of the introduction in a slightly better editorial version- too much disjointedness into bullets, listing and plain text. I think the duration of the waves can be removed are in chapter 2.1
· Please pardon me, but pointing out potential counseling requests for patients at age 0 sounds rather sarcastic (line 89).
Author Response
Dear Reviewer,
we thank you for your considerations and for the suggestion of changes that will certainly be useful to improve both the quality of our manuscript and our future studies. Below we list the responses to your reviews, which we hope can be considered satisfactory by you.
Major:
- Is it a cross-sectional study design?
As you have rightly pointed out, we did not specify that our study have a cross-sectional design. We have therefore included it in the title and in the Method section of the manuscript.
- Chapter 2.1 is far too short - please expand with detailed descriptions of data acquisition methods
As you requested, we have expanded the section of data recruitment with a more detailed description of the data acquisition methods.
Minor:
- I would suggest presenting the last paragraph of the introduction in a slightly better editorial version- too much disjointedness into bullets, listing and plain text. I think the duration of the waves can be removed are in chapter 2.1
As you requested, we have presented the last paragraph of the introduction in an improved editorial version and removed the duration of the waves in chapter 2.1.
- Please pardon me, but pointing out potential counseling requests for patients at age 0 sounds rather sarcastic (line 89).
We adjusted the age range of the enrolled patients in the line 89.
Best regards,
Emilia Matera
Reviewer 2 Report
The authors investigated an interesting manuscript on the adolescent mental health during COVID-19 pandemic based on a single-institutional data.
1) Title: it seems redundant, as the University Hospital of Bari, Italy is in the title. Is the University Hospital of Bari, Italy famous? Please remove as possible.
2) Abstract: The abstract is so descriptive, and there is no number of total enrolled patients or statistical significant number in the results of abstract section. Please add the definite numbers to improve the statistical significance in your data.
3) Introduction (line 51-59) is too long and redundant and the context seems not to be directly associated with COVID19 mental health change. Please shorten those sentences or remove.
4) Introduction (line 65-68): the reference that you mentioned is only in Canada. And your almost references are from European or US, not from Asia (Otherwise a few China). How are you sure that the mental change in adolescents during COVID19 is worldwide? Please add Asian data. (for instance: J Pers Med. 2022 Apr 4;12(4):576.)
5) Figure 1 and Figure 2: the letter characters are too small to recognize them. Please add the larger-point characters in Figures.
6) Discussion: (line 207-221): the authors mentioned “somatization symptoms”. However, it is unclear what diseases are in the definition. Please more discuss the somatic symptoms during COVID compared to prior. (There may be some relevant gastrointestinal symptoms in the pandemic. (The possible reference is J Pers Med. 2022 Jul 14;12(7):1144.)
Author Response
Dear reviewer,
we thank you for your considerations which will certainly be useful in improving the scientific quality of our manuscript. We list the responses to your reviews below.
1) Title: it seems redundant, as the University Hospital of Bari, Italy is in the title. Is the University Hospital of Bari, Italy famous? Please remove as possible.
As you have suggested, in the title we removed the “University Hospital of Bari, Italy” and we replaced it with the term “Italian”.
2) Abstract: The abstract is so descriptive, and there is no number of total enrolled patients or statistical significant number in the results of abstract section. Please add the definite numbers to improve the statistical significance in your data.
As you have rightly suggested, in the abstract we included the total number of the enrolled patients and the statistical significant numbers in the results section.
3) Introduction (line 51-59) is too long and redundant and the context seems not to be directly associated with COVID19 mental health change. Please shorten those sentences or remove.
In the Introduction (line 51-59) we shortened the redundant sentences not directly associated with COVID19 mental health changes.
4) Introduction (line 65-68): the reference that you mentioned is only in Canada. And your almost references are from European or US, not from Asia (Otherwise a few China). How are you sure that the mental change in adolescents during COVID19 is worldwide? Please add Asian data (for instance: J Pers Med. 2022 Apr 4;12(4):576.)
Sharing your consideration, we included in the Introduction section of the manuscript Asian data.
5) Figure 1 and Figure 2: the letter characters are too small to recognize them. Please add the larger-point characters in Figures.
We added the larger-point characters in the figures.
6) Discussion: (line 207-221): the authors mentioned “somatization symptoms”. However, it is unclear what diseases are in the definition. Please more discuss the somatic symptoms during COVID compared to prior. (There may be some relevant gastrointestinal symptoms in the pandemic. (The possible reference is J Pers Med. 2022 Jul 14;12(7):1144.)
As you rightly suggested, in the Discussion we specified the somatic symptoms during COVID compared to prior and discussed them using the suggested reference.
Kind regards,
Emilia Matera
Reviewer 3 Report
Major comments:
The study aims to present the ongoing mental impact of the pandemic; however, the data collection is based on emergency room visits. These are very specific events. It is not surprising that the number of visits and referrals to ER due to mental health problems is greater during the time of pandemic, this is to be expected in such a crisis. But this does not mean that these consequences are long-lasting. To show long-term consequences – you need to investigate whether the trend continues beyond the critical period of the pandemic, but we are not there yet. This type of study requires prospective follow-up, and this is not the case of the present study. It is presumptuous to conclude from a retrospective study based only on the documentation of visits to the emergency room that there are longitudinal mental implications of the pandemic.
The Introduction section is not sufficient. The information found in it does not provide enough answers about what is already known and what is the gap that this research is trying to fill. The psychological implications on adolescents are well known, but it is unclear what is the situation in Italy compared to the rest of the world? What led you to investigate this topic?
The Methods chapter must be elaborated and divided into sections: Design, Participants, Procedure, Setting, Data analysis.
The statistical tests used are very basic with very little implications. They can show only descriptive analyses and differences between two groups.
Minor comments:
Introduction:
The first sentence – lines 33-34 is relatively controversial, there are evidence demonstrating similar fatality rated of Influenza and COVID-19. Since this statement is not at all related to the essence of your research, I recommend giving it up and starting with another opening sentence.
Results:
Table 1: There is no need to present the values of the statistical tests performed, means, SD / frequencies, and p-values are enough.
p-value that includes "E" means there was a mix up in the statistics or it is not significant. Either way – it is not clear to the readers and needs to be modified.
The labeling of the statistical tests is too long, you can leave is as "chi square" or "Pearson", it is more friendly to the readers.
Discussion chapter – start with your major findings rather than general statement.
Author Response
Dear Reviewer,
we thank you for your time spent reviewing our manuscript and for the suggestion of changes that will certainly be useful to improve both the quality of our manuscript and our future studies. Below we list the responses to your reviews, which we hope can be considered satisfactory by you.
Major comments:
The study aims to present the ongoing mental impact of the pandemic; however, the data collection is based on emergency room visits. These are very specific events. It is not surprising that the number of visits and referrals to ER due to mental health problems is greater during the time of pandemic, this is to be expected in such a crisis. But this does not mean that these consequences are long-lasting. To show long-term consequences – you need to investigate whether the trend continues beyond the critical period of the pandemic, but we are not there yet. This type of study requires prospective follow-up, and this is not the case of the present study. It is presumptuous to conclude from a retrospective study based only on the documentation of visits to the emergency room that there are longitudinal mental implications of the pandemic.
The Introduction section is not sufficient. The information found in it does not provide enough answers about what is already known and what is the gap that this research is trying to fill. The psychological implications on adolescents are well known, but it is unclear what is the situation in Italy compared to the rest of the world? What led you to investigate this topic?
Thanks for your first and second major comments. We agree with your right considerations. Our study aims present the ongoing mental impact of the pandemic. Although during the time of pandemic, it is expected that the number of visits and referrals to ER due to mental health problems is greater, our hypothesis was that the persistence of the stressful event (pandemic) has determined consequences on the psychopathological conditions of young people in terms of type and severity, in particular between the COVID 19 first and second pandemic waves, as emerged from the obtained results. We are aware that the aforementioned consequences cannot be interpreted as long-lasting. For such reasons we have made changes in the Title, Introduction, Discussion and Conclusion of the manuscript to make the terminology we use less ambiguous and add more detailed informations about the topic of our study and the literature gap it aims to fill.
The Methods chapter must be elaborated and divided into sections: Design, Participants, Procedure, Setting, Data analysis.
As you suggested, we elaborated and divided the Method chapter into section Design, Participants, Procedure, Setting, Data analysis.
The statistical tests used are very basic with very little implications. They can show only descriptive analyses and differences between two groups.
As you correctly stated, we are aware that our statistical analysis is simple and with little implications. Given the important consequences of the pandemic on the mental health of children and adolescents, it seemed useful to share these preliminary results of our research, albeit mostly descriptive. However, we hope our results are in any case clear and interesting for the readers, as well as consistent with the aims of our study.
Minor comments:
Introduction:
The first sentence – lines 33-34 is relatively controversial, there are evidence demonstrating similar fatality rated of Influenza and COVID-19. Since this statement is not at all related to the essence of your research, I recommend giving it up and starting with another opening sentence.
In consideration of your exact objection, we changed the introductory sentence.
Results:
Table 1: There is no need to present the values of the statistical tests performed, means, SD / frequencies, and p-values are enough.
p-value that includes "E" means there was a mix up in the statistics or it is not significant. Either way – it is not clear to the readers and needs to be modified.
In line with both your suggestions, we revised the entire table trying to make it more reading friendly. We deleted the columns of the statistical test results, modified the tables and adjusted the values removing “E” that was an excel refuse.
The labeling of the statistical tests is too long, you can leave is as "chi square" or "Pearson", it is more friendly to the readers.
According with this useful suggestion, we revised the entire paragraphs of statistical analysis and results synthetizing the names of tests.
Discussion chapter – start with your major findings rather than general statement.
Thanks for your final suggestion. We changed the initial section of the Discussion chapter starting with our major findings.
Best regards,
Emilia Matera
Round 2
Reviewer 1 Report
Dear Authors,
Thank you for providing corrections. Although some of them have been completed, it is the core of the study continues to remain at a low level. It does not find the novelty that the paper could bring to the world of science.
Author Response
Dear reviewer,
we thank you again for your availability. We are very sorry if the core of the study continues to remain at a low level. In order to better clarify the novelty that the paper could bring to the world of science, we integrated the introduction section of the manuscript with additional information and references which highlight the current state of the art on this topic. We hope our changes will be satisfactory for you.
Best regards,
Emilia Matera
Reviewer 2 Report
Excellent work.
Author Response
Dear reviewer,
we thank you for appreciating our work.
Best regards,
Emilia Matera
Reviewer 3 Report
I thank the authors for their comments, the manuscript was significantly improved
Author Response
Dear reviewer,
we thank you very much for your approval.
Best regards,
Emilia Matera